# SchoolAIR: A Citizen Science IoT Framework Using Low-Cost Sensing for Indoor Air Quality Management

**DOI:** 10.3390/s24010148

**Published:** 2023-12-27

**Authors:** Nelson Barros, Pedro Sobral, Rui S. Moreira, João Vargas, Ana Fonseca, Isabel Abreu, Maria Simas Guerreiro

**Affiliations:** 1FP-I3ID—Fernando Pessoa Institute for Research, Innovation and Development, 4249-004 Porto, Portugal; afonseca@ufp.edu.pt (A.F.); iabreu@ufp.edu.pt (I.A.); mariajoao@ufp.edu.pt (M.S.G.); 2CINTESIS.UFP—Center for Health Technology and Services Research, 4200-450 Porto, Portugal; 3LIACC—Artificial Intelligence and Computer Science Laboratory, University of Porto, 4200-465 Porto, Portugal; pmsobral@ufp.edu.pt (P.S.); rmoreira@ufp.edu.pt (R.S.M.); 4Faculty of Science and Technology, University Fernando Pessoa, 4249-004 Porto, Portugal; 36431@ufp.edu.pt

**Keywords:** low-cost sensors, scalable IoT sensing system, indoor air quality, ventilation management, citizen science, schools

## Abstract

Indoor air quality (IAQ) problems in school environments are very common and have significant impacts on students’ performance, development and health. Indoor air conditions depend on the adopted ventilation practices, which in Mediterranean countries are essentially based on natural ventilation controlled through manual window opening. Citizen science projects directed to school communities are effective strategies to promote awareness and knowledge acquirement on IAQ and adequate ventilation management. Our multidisciplinary research team has developed a framework—SchoolAIR—based on low-cost sensors and a scalable IoT system architecture to support the improvement of IAQ in schools. The SchoolAIR framework is based on do-it-yourself sensors that continuously monitor air temperature, relative humidity, concentrations of carbon dioxide and particulate matter in school environments. The framework was tested in the classrooms of University Fernando Pessoa, and its deployment and proof of concept took place in a high school in the north of Portugal. The results obtained reveal that CO_2_ concentrations frequently exceed reference values during classes, and that higher concentrations of particulate matter in the outdoor air affect IAQ. These results highlight the importance of real-time monitoring of IAQ and outdoor air pollution levels to support decision-making in ventilation management and assure adequate IAQ. The proposed approach encourages the transfer of scientific knowledge from universities to society in a dynamic and active process of social responsibility based on a citizen science approach, promoting scientific literacy of the younger generation and enhancing healthier, resilient and sustainable indoor environments.

## 1. Introduction

The COVID-19 pandemic has changed public perception of the indoor air quality (IAQ) in non-residential buildings. The adequate management of IAQ in schools is of particular relevance, given the vulnerability of the young occupants and the daily amount of time spent indoors [1].

Ventilation practices and outdoor pollution have been proven to have a relevant impact on the indoor air quality of schools in urban environments [2,3].

In the Mediterranean area, most of the school buildings lack mechanical ventilation systems and IAQ management relies on natural ventilation systems based on air permeability of the building envelope and manual airing of the classrooms through window or door opening [4,5,6]. However, natural ventilation systems may not be enough to guarantee adequate IAQ [7,8]. In a study carried out in schools with natural ventilation in Barcelona (Spain) and Cassino (Italy), Pacitto et al. [9] demonstrate the risk associated with exposure to particles, highlighting the impact of this type of pollution on students’ health.

Frequently dependent on the initiative of users to open windows or doors, effective natural ventilation requires these users to be aware of IAQ constraints and to have the necessary knowledge to act accordingly in different scenarios of indoor and outdoor conditions [10,11].

The relevance of this topic is that low IAQ in educational buildings is widely recognized to have a negative impact on students’ performance, development and health [1]. Climate change impacts are expected to enhance IAQ related problems, namely due to increased levels of particulate matter and photochemical pollutants in outdoor air, and also due to alterations in ventilation practices in the adaptation to higher temperatures and relative humidity levels [12,13].

The use of real-time sensors and IoT to monitor IAQ parameters is widely spread among the scientific community [14,15,16,17]. Low-cost sensors have a high potential in environmental monitoring applications, providing data with a high temporal and spatial resolution at affordable costs. The most common IAQ parameters monitored with real-time low-cost sensors are indoor air temperature (T) and relative humidity (RH), and the concentration of air pollutants like carbon dioxide (CO_2_), particulate matter (PM), carbon monoxide (CO), nitrogen dioxide (NO_2_), volatile organic compounds (VOCs) and ozone (O_3_).

To build sustainable and resilient communities, as depicted in the Sustainable Development Goals (SDGs) of the United Nations 2030 Agenda [18], it is essential to involve different actors in the society. Raising awareness, especially in the younger generations, is of the utmost importance to align perceptions, attitudes, and behaviours with the principles of Sustainable Development. Citizen science (CS) initiatives are defined as the involvement of communities and individuals in scientific experiments to explore issues concerning to them, with minimum involvement of professional scientists [19]. CS has social and educational impacts by increasing literacy, producing knowledge, and enhancing behavioural change through citizen engagement and participation in decision-making processes supported by scientific evidence.

By involving citizens in projects that address sustainability problems, CS initiatives have the potential to promote sustainability transitions [20] and to make relevant contributions to the achievement of the SDGs, specifically SDG 3: Good Health and Wellbeing, SDG 11: Sustainable Cities and Communities, and SDG 13: Climate Action [21].

It is common practice in CS projects to use digital technologies to promote interactions and collect data. The concept of citizen-sourcing has emerged as a joint application of CS and big data collection, with high potential value for citizens, researchers, policymakers, and the society as a whole [22].

In what concerns air quality monitoring, combining the use of low-cost sensors with a citizen science approach enlarges the scope of monitoring campaigns and, at the same time, enhances public awareness, engagement, and scientific literacy regarding this important issue. In this scope, the following examples of recent CS projects were found in the literature:iSCAPE: active between 2016 and 2019, this project used low-cost sensors aiming to improve the smart control of air pollution in European cities [23];hackAIR: this project took place from 2016 to 2018 and aimed to enable citizens to use do-it-yourself sensors to measure the quality of the air they breathe. An open platform was launched to map air quality across Europe [24];CanAirIO and UNLOQUER: two citizen science initiatives that took place in Colombia to monitor air urban pollution levels using low-cost sensors for PM [25];Lu et al. [26] used low-cost sensors in a CS initiative aiming to monitor PM levels in outdoor air in Southern California;Soc-IoT: an open-source and citizen-centric IoT framework combining real-time environmental sensors with a visualization application designed for indoor and outdoor environmental monitoring [27];SOCIO-BEE: currently active, this project uses wearable modules for city-scale air pollution monitoring campaigns [28].

School communities are particularly receptive to CS projects, specially when these enable the creation of scientific experiments for students and their involvement in decision-making processes regarding the improvement of environmental conditions [29]. In a recent literature review focusing CS initiatives in school contexts, Solé et al. [30] highlight the importance of including formal education objectives in CS projects to fully achieve the development of students’ skills. Different examples of CS-based studies focusing on air quality and students’ exposure to air contaminants can be found in the literature:Grossberndt et al. [31] coordinated CS projects in two Norwegian high schools, where students were asked to develop their own research project on the topic of air quality. Low-cost sensors were used to measure air pollutants in different urban spots, in the neighbourhood of the school, and also indoors;Varaden et al. [32] describe a CS initiative that took place in five London primary schools. The level of air pollution exposure during their daily route to and from school was measured using outdoor air quality sensors incorporated in students’ backpacks;Ellenburg et al. [33] report the results of the AQTrecks project, which involved the use of low-cost air quality sensors to measure carbon monoxide (CO), carbon dioxide (CO_2_) and particulate matter (PM). Personal air monitors and a smartphone app were developed specifically to monitor both indoor and outdoor air in school contexts, covering 95 schools across the United States of America.Ulpiani et al. [34] describe a CS project involving Australian schools—The Schools Weather and Air Quality (SWAQ) network—aiming to study the role of urbanization on outdoor environmental quality. This network collects data regarding the concentration of sulfur dioxide, nitrogen dioxide, carbon monoxide, ozone, $PM_{10}$ and $PM_{2.5}$.

However, there seems to be a research gap in the use of CS initiatives focusing on indoor air quality and ventilation management, and covering this topic in school contexts could have a relevant impact on students’ health and performance. In this scope, the main objective of the present research is to design, validate and operationalize a scalable IoT system architecture to support ventilation management and, thus, improve IAQ in schools through the use of low-cost sensors and IoT—the SchoolAIR framework.

The SchoolAIR framework aims also to be highly scalable to support a large number of monitored classrooms in country-size spreading schools, along with fault tolerance and data security.

This framework simultaneously promotes the development of technical skills, since the monitoring system is based on Do-It-Yourself (DIY) sensors, which are expected to be assembled by students. Students’ engagement in this citizen science project is expected to raise awareness regarding the impact of the environment on health, and disseminates good practices regarding ventilation procedures. This is particularly relevant in climate change adaptation scenarios, contributing to the resilience and sustainability of school communities, which will be spread to families and surrounding communities.

The remainder of this paper is organized as follows: Section 2 presents the SchoolAIR framework, describing the selected sensors and the hardware and software architecture as well as the preliminary tests that took place in university classrooms to validate the system; Section 3 focuses on the deployment of the SchoolAIR framework in a school located in the northern region of Portugal; the paper closes with the discussion and conclusions section.

## 2. Development of the SchoolAIR Framework

The SchoolAIR framework is based on an IoT system that aims to collect and aggregate data regarding IAQ parameters from various classrooms in different schools, through the use of easy DIY sensor Edge-Nodes/Motes. Based on a citizen science approach, the framework was developed under the premise that the school community would be autonomously involved in the assembling and deployment of the system nodes.

### 2.1. SchoolAIR Architecture Overview

The SchoolAIR architecture was conceived with scalablility, fault tolerance and security concerns in mind, to be able to efficiently support the country-wide monitoring of school indoor air environmental data, using a common design for large-scale IoT systems [35]. The SchoolAIR architecture can scale to a large number of sites using cloud elasticity. It is also fault-tolerant since each node buffers the sensor data in local storage with periodic backups to the Google Cloud. Thus, in the event of a temporary network disconnection or hardware/software failure, the system can recover and resume its operation in a safe state. The SchoolAIR architecture also assures the integrity and security of all communications among the nodes. They are performed over encrypted channels (SSL) across the Internet and using WPA2 Enterprise encryption with an exclusive VLAN tag on the school Wi-Fi network.

The SchoolAIR architecture, presented in Figure 1, is structured into three layers: (i) central cloud instance (Cloud-Node) that centrally aggregates all collected data and provides dashboard facilities; (ii) local in-school-premises Fog-Node instances, responsible for locally aggregating data collected from various classrooms Edge-Nodes; also providing the means for depicting and analysing local data-sets; (iii) low-cost DIY monitoring local Edge-Nodes/Motes installed in each classroom, responsible for collecting environmental data from classrooms; Edge-Nodes assemble various environmental sensors and are socket powered in the classrooms. The data collected by these monitoring Motes is sent via WiFi to a local school Fog-Node instance. Each Fog-Node instance locally stores data from each Edge-Node and depicts it in the local school dashboard. Fog-Nodes are also responsible for sending collected data to the central Cloud-Node instance, which aggregates data-sets from all monitored schools, thus providing a system-wide real-time overview of the entire collected data.

#### 2.1.1. Cloud-Node

The central Cloud-Node component is based on the HomeAssistant (HA) open-source platform (version 2023.2.1), designed to integrate and manage various types of smart devices in generically any home or building environment. The HA has native support for various operating systems on different platforms, thus being able to be deployed on *Linux* and *Windows* machines, among other options. The HA premise is data privacy, since it has been designed to locally keep aggregated data from connected devices.

Although HA focuses on home automation, several factors led it to be chosen as the base system for the SchoolAIR project. The HA was designed to receive and communicate with a large number of devices simultaneously, while offering advanced security features such as encrypted communications.

The HA platform provides also *Integrations*, which are pre-configured packages that can be easily installed and integrated to provide additional functionalities and thus expanding the capabilities of the base system. In order to take advantage of this potential, several *Integrations* were used to improve the security and robustness of the Cloud and Fog-Node instances, allowing easy configuration and maintenance of the Edge-Node monitoring Motes and improving the way to visualise all the information collected over time. The set of *Integrations* used in the SchoolAIR framework is presented in Figure 2.

#### 2.1.2. Fog-Node

Every school Fog-Node instance stores data in local database and provides also a local dashboard for depicting and analysing all recorded information collected over time. In addition, data is also sent periodically to a central Cloud-Node instance, identifying not only the classroom but also the school to which it belongs. Both local Fog-Nodes and central Cloud instances provide the means for storing, depicting and analysing the collected data-sets by authorised users only. The Fog-Nodes were implemented with *Raspberry Pi* boards, each responsible for running a local HA instance.

To improve scalability, the system is organized into levels to efficiently accommodate several country-wide schools. Each school deploys a Fog-Node instance for aggregating several classroom Edge-Nodes. All communications between the Cloud-Node and Fog-Nodes, and Fog-Node and Edge-Nodes, are secured by state-of-the-art encrypted channels. Furthermore, each local Fog-Node instance implements backup mechanisms for improving fault-tolerance.

#### 2.1.3. Edge-Node

The Edge-Nodes are based on a microcontroller for collecting data from various sensors and sending them to local Fog-Nodes. The ESP32 board, visible in Figure 3, was chosen due to its processing power, connection capabilities and flexibility for supporting various types of sensors. This particular model provides also an U.FL socket for an external antenna that improves WiFi signal range.

Furthermore, the monitoring Edge-Nodes (aka Motes) use *EspHome* (version 2022.12.8), an open-source platform designed to facilitate the integration and control of ESP32 and ESP8266 microcontrollers. This platform makes it easier to configure these microcontrollers, through intuitive YAML files, to customise their behaviour and assembly of various sensors. *EspHome* eliminates the need to use low level programming of embedded platforms. In addition, it allows *Over-The-Air* (OTA) updates, thus making Edge-Nodes more easy to install, maintain and update. Figure 4 shows an example configuration of one monitoring Mote, containing the sensor types and sensor names that will associated with the digital-twin entities in the local Fog-Node HA instance. The configuration contains also the microcontroller pins to be used and the data collection triggering periods for each of the sensors. The first configuration of the microcontroller needs to be carried out via cable, however, afterwards all updates can be performed *Over-The-Air*.

Sensors play a crucial role in this system, as they make it possible to capture real time information about the environment in each monitored classroom. The system measures CO_2_ concentrations, particulate matter ($PM_{10}$, $PM_{2.5}$ and $PM_{1}$), as well as complementary IAQ parameters such as relative humidity and air temperature. The sensors used in our local Edge-Nodes are presented in Figure 5a–c.

Carbon Dioxide (CO_2_) plays a fundamental role in classroom monitoring, since CO_2_ levels are directly related to the occupancy, level of ventilation and indoor air quality of the space, and high concentration may have an effect on students’ well-being, performance and health.

The MH-Z19 sensor (see Figure 5a), uses the principle of non-dispersive infrared detection (NDIR) to measure CO_2_ levels by emitting infrared light with a certain frequency and then measures the amount of light absorbed by CO_2_ present in the air. Based on this light absorption, the sensor determines the concentration of CO_2_ present in the space. This model is typically used in indoor air quality monitoring projects, as well as in smart home projects, due to its low error rate and long lifetime [36].

The presence of particulate matter (PM) in indoor air is related with several causes/sources. For example, the movement of people, the release of particles into the air when breathing, talking or coughing. Also, several activities being carried out in the indoor environment may lead to the release of PM to air—cleaning and dusting, smoking, cooking, gardening, among many others. Outdoor air is also a relevant source of PM in indoor air since these particles penetrate indoor environments through windows, doors, and forced ventilation systems. Indoor air contamination with PM is widely recognized as a health hazard, particularly in school environments [9]. Gathering information about the concentration of PM is therefore also essential as a good proxy for both indoor contamination and contamination from outdoor air used for ventilation, particularly in urban environments.

The PMS5003 sensor [37], presented in Figure 5b, measures the concentration of particles suspended in the air, using an optical laser scattering system. The sensor emits a laser beam and measures the amount of light scattered by the particles to determine the particle concentration. It is capable of measuring particles with an aerodynamic equivalent diameter of 10 µm ($PM_{10}$), 2.5 µm ($PM_{2.5}$) and 1 µm ($PM_{1}$), thus allowing the measurement of particles of dust and smoke, among others.

Relative humidity (cf. a ratio, expressed in percent, of the amount of atmospheric moisture present relative to the amount that would be present if the air was saturated) can, at too low levels, contribute to eye irritation, dry throat or respiratory discomfort, while high levels can contribute to the appearance of bacteria and condensation in the space. This is also an important element to monitor, as it influences people’s comfort in a classroom and can directly affect attendance and focus.

Air temperature (T) is also an essential factor for thermal well-being in enclosed spaces. Inadequate temperature levels can contribute to feelings of discomfort, fatigue and concentration difficulties [1,2]. Monitoring this element is also significant because in occupied enclosed spaces, the temperature tends to rise with time without acclimatization countermeasures.

The DHT22 sensor [38], presented in Figure 5c, is a temperature and relative humidity sensor with good accuracy. Having small dimensions and low energy consumption, it is widely used in automation projects and meteorological stations. Table 1 contains information about the dimensions, measurement range, accuracy and response time of the three sensors used in the SchoolAIR framework.

### 2.2. SchoolAIR Data Flow

The system’s main purpose is collecting data over time and storing sensors’ time series both at local Fog and central Cloud Nodes. For this purpose, the HA *InfluxDB* integration (version 4.5.0) was used to provide database support for storing and querying time series. This HA integration provides efficient and scalable support for handling large volumes of data, which is vital in this project, given the amount of information to be aggregated and manipulated.

The HA *Grafana* integration (version 8.1.0) was also used to provide easy and intuitive data visualisation facilities. This HA integration provides numerous types of data visualisations possibilities and time filters usages.

Another fundamental requirement was to ensure collected data is not lost even when Fog or Cloud Nodes suffer from malfunctions or crashes. For this purpose, the HA *Google Drive Backup* integration (version 0.110.1) was used, allowing backup copies of the entire system to be stored on *Drive*. These copies are performed automatically, on a daily basis. Therefore, such replication strategy provides easy way for supporting data recovering from all HA instances.

Finally, the HA *NGINX* integration (version 3.5.0) was used in the central Cloud-Node instance, since it is exposed on the Internet. This HA integration provides an intermediary node between clients and the central Cloud-Node instance, thus adding a layer of security, since it provides SSL/TLS communications’ encryption thus protecting data exchanges between nodes.

The information flow originates from schools’ Edge-Nodes that collect sensor info every 5 min and communicate it to the local Fog-Node instance. The connection is provided over WiFi, since most country’s schools possess structured wireless infrastructures. The collected data is stored locally and may be accessed in real time.

The HA was tailored for running locally, therefore several options had to be explored to allow HA instances to communicate with each other. One of the considered options was the use of MQTT, which provides a broker for handling messages exchanged between publishing and consuming HA instances. This solution was complex to configure given the number of sensors involved. Therefore, it would not scale as well as other available options. Another option was the use of HA *Remote HomeAssistant* integration, which allows HA instances to communicate with each other remotely. However, all HA Fog and Cloud Nodes instances needed to be exposed on Internet. This was not possible since Portuguese school’s WiFi infrastructures have high level security rules against port forwarding, therefore preventing this integration. The alternative solution was to instantiate a central HA Node, available at the address https://airmon.ufp.pt (accessed on 1 October 2023). The communication with this central HA instance is one way, i.e., only the local Fog-Node instances can send info to the central Cloud instance. The communication process is carried out by configuring an automation in each school Fog-Node instance. As presented in Figure 6, every 3 min, each Fog-Node dynamically executes a script for collecting all data from local *EspHome* integrations, i.e., from all sensors deployed in all Edge-Nodes installed in the classrooms. The script also sends the collected data via a *REST command* provided by the *Application Programming Interface* (API), exposed by the central Cloud-Node instance.

Each school has an authentication *token* with a long-life period, allowing communication between instances over a long time. In order to organise the received data, a tagging nomenclature was implemented for naming each sensor, i.e., identifying the school, classroom and sensor type (cf. “School Name—Room Name—Sensor Type”). This process is depicted in Figure 7.

The configuration process is performed once in each school Fog-Node instance. Local Edge-Nodes sensors may be added without any configuration needs, since the system has the ability to automatically detect them and send their data to the central Cloud instance. When the central instance receives data from a new sensor, it automatically creates its records and stores them in the database. This whole mechanism is guaranteed by the HA facilities thus allowing new Edge-Nodes sensors and Fog-Nodes to be added easily.

The *Grafana* integration allows the extraction of all information collected by the various sensors into a file in the *Comma Separated Values* (CSV) format. Authorised users need only to select the sensors and time period to be extracted and exported. The CSV files are later processed by Python *scripts* to produce data-sets to be consumed by Machine Learning (ML) pipelines.

### 2.3. Preliminary Tests

As a preliminary test, the developed framework and associated IoT system were applied in classrooms of University Fernando Pessoa, in the city of Porto (41°10′22″ N, 8°36′40″ W). The university building has a total area of 5500 m^2^ throughout five floors with classrooms, laboratories, and other facilities such as a library, an auditorium, and a bar. The 29 classrooms show a total area of 1190 m^2^.

The first step was to use an inter-comparison procedure aiming to verify sensors’ precision to maximize data quality and assure inter-comparison of results [39,40]. Sensors may be rejected based on these results. As an example of the procedure, three CO_2_ sensors were placed side by side in a classroom, under normal operation conditions. Figure 8 shows the CO_2_ concentration for the three CO_2_ sensors in a classroom during one week in March 2023. The results show that sensor 1 does not follow the same values as sensors 2 and 3.

In addition, as the correlation plot for the CO_2_ sensors revealed a different sensitivity for sensor 1 (Figure 9), the non-parametric Friedman test was applied. The Friedman test showed a statistically significant difference in CO_2_ concentration depending on the sensor used (*p* < 0.001). Post hoc analysis with a Bonferroni correction revealed no significant differences between sensor 2 and sensor 3 measurements (*p* = 0.594), but a statistically significant difference between the sensor 1 and both sensors 2 and 3 measurements (*p* < 0.001). As an example of the importance of this procedure, the statistical analysis lead to the replacement of a sensor—sensor 1. The same procedure was followed for the other sensors, covering all the analyzed parameters, and no other sensors had to be replaced.

The preliminary tests included the set up of the sensors in three representative classrooms at University Fernando Pessoa to measure environmental parameters during the normal operation of teaching activities: classroom 106, on the first floor, with 47.0 m^2^, 3.0 m of ceiling height and 48 seats, with no ventilation apart from the air transfer through the entrance door and a small shutter-type window of 0.4 m^2^; classroom 204, on the second floor, with 34.0 m^2^, 3.0 m of ceiling height and 27 seats, with natural ventilation through two free-opening tilt-and-turn windows of 1.8 m^2^ each. Both rooms face east. The third room, classroom 210, also on the second floor, has 36.0 m^2^, 3.0 m of ceiling height and 27 seats. This classroom has natural ventilation through two free-opening tilt-and-turn windows of 1.8 m^2^ each. Unlike the previous rooms, this one faces west towards the prevailing NW winds. Data from the monitoring Motes measuring the concentration of CO_2_, $PM_{10}$, $PM_{2.5}$ and $PM_{1}$ in indoor air, and also the environmental parameters T and RH (Figure 10) was collected continuously and registered every five minutes.

The dashboard presented in Figure 10 aims to present an overview of the monitored parameters in real-time. This dashboard allows, with a single click, to visualize a more detailed plot for each of the parameters, as shown in Figure 11.

Sensors’ measurements exhibited consistency with scholar activities—the rise and fall of CO_2_ concentrations correspond to classes on the five working days of the week. International indoor air quality guidelines indicate 1000 ppm as the reference value for CO_2_ concentration, and consider 2000 ppm as the intervention value for this pollutant [41]. The results obtained in the classrooms of University Fernando Pessoa show high levels of CO_2_ concentration during classes, highlighting the importance of continuously monitoring CO_2_ levels to support decisions regarding ventilation practices (e.g., opening windows) for IAQ improvement.

## 3. Deployment of SchoolAIR in Alpendorada High School

For concept trial and demonstration purposes, the deployment of the SchoolAIR framework was performed in the Alpendorada High School (41°5′0″ N, 8°14′35″ W), in the northern part of Portugal. The city of Alpendorada shows a Mediterranean climate—Csb, with mild wet winters and warm dry summers, according to Köppen’s classification.

This project was launched with several technical meetings between the UFP’s research and development team and two resident teachers at Alpendorada High School. These meetings were crucial for the teachers to become acquainted with the project’s goals, and for the teachers and the research team to acknowledge technical deployment specificities. Sensors location was assessed in a ulterior visit to the framework deployment site.

### 3.1. Local Nodes Assembly and Configuration

The high school community received a detailed assembling and configuration manual for both the local SchoolAIR Edge and Fog-Nodes. This manual contained plain and straightforward instructions together with default (or examples) configuration files for the modules of the system. The instructions’ manual allowed the school community (cf. local school teachers and their students) to locally assemble and deploy all system nodes autonomously, and thus develop students’ technical skills.

#### 3.1.1. Deployment of Local Monitoring Nodes in Classrooms

The Alpendorada high school building has two floors and a total area of 6480 m^2^. The prototypes were assembled in two classrooms (Figure 12a). Both classrooms are on the first floor and partially face the prevailing NW winds. Classroom 12 has 44.8 m^2^ and 28 seats. Classroom 13 has 49.0 m^2^ and also 28 seats. Both have 3.0 m ceilings and natural ventilation with four free-opening tilt-and-turn windows of 0.72 m^2^ each. Since the SchoolAIR framework aims to support decision-making in ventilation management, it is of the utmost importance to obtain data regarding outdoor conditions. For this purpose, a monitoring node has also been installed in outdoor environment (Figure 12b). The outdoor monitoring location is sheltered from the rain, and faces the prevailing NW winds. The partner teachers and students were able to assemble and deploy the Edge-Nodes and respective sensors in all selected locations.

#### 3.1.2. Sensors Inter-Comparison

All monitoring Motes were placed in the same classroom for a week. The procedure proposed in the preliminary tests phase was followed for sensors inter-comparison. The readings of the three CO_2_ sensors follow each other (Figure 13).

The correlation plot (Figure 14) reinforced sensors selection. Based on these results, no sensors were rejected.

### 3.2. Outputs

Data from the monitoring Motes (CO_2_, particles $PM_{10}$, $PM_{2.5}$ and $PM_{1}$, T and RH) in each classroom and outdoors (Figure 15) was collected continuously and registered every five minutes.

As observed in the preliminary tests in University Fernando Pessoa, the measurements regarding CO_2_ concentration are consistent with scholar activities, and show that the reference value of 1000 ppm [41] is frequently exceeded during classes in Alpendorada High School. Figure 15 also shows the dynamic behaviour of PM concentrations in indoor and outdoor environments, indicating that outdoor air is a relevant source of contamination regarding this pollutant.

## 4. Discussion and Conclusions

IAQ within school premises poses significant challenges, hampering the teaching/learning environment. The main problem in this context is one of perception. Since the main environmental parameters that generate poor IAQ are odorless and invisible, this situation is often overlooked. It is essential to have an adequate monitoring system that can promptly alert those overseeing these spaces and empower them to take proactive measures to enhance indoor air quality across various scenarios.

The preliminary tests and the deployment of the SchoolAIR framework in Alpendorada high school show that this system is fully developed and ready to be implemented. The simplicity of assembling, configuring and installing the SchoolAIR framework makes it especially suitable for implementation in school environments where locally available technical skills are limited. Thus, the SchoolAIR framework is a valuable tool to be used in CS projects focusing IAQ in schools, involving teachers and students in the assembling of the system and in the IAQ monitoring process.

The results obtained so far show the importance of having an easy-to-use tool to continuously monitor IAQ conditions and thus detecting poor environmental conditions that need to be addressed:Exceeding the reference values regarding CO_2_ concentration occurs frequently during classes. This IAQ problem is frequent in school environments, reported by several studies (e.g., [1,2,3]). These high values of CO_2_ indicate the need to introduce fresh air into the classroom through ventilation—either opening windows or doors in manual ventilation systems, or increasing the air flow if mechanical ventilation systems are being used. The SchoolAIR framework enables the detection of ventilation needs in real-time, thus reducing students’ exposure to inadequate IAQ conditions. Students can easily be involved in this process, promoting a citizen science approach that will raise awareness of the entire school community to IAQ problems and the importance of adequately managing ventilation;The influence of outdoor air conditions in IAQ is also highlighted in the results obtained in Alpendorada High School regarding the pollutant PM. In fact, by detecting in real time an increase in outdoor air pollution levels, the SchoolAIR framework enables the adoption of measures to minimize the contamination of indoor air with outdoor pollutants—closing windows and doors, and turning off ventilation systems that are fed with outdoor air. Specifically in what concerns PM, climate change is expected to increase the frequency of high concentration episodes, and thus it is of the utmost importance to adopt timely protective measures that may have significant impacts in the health of the entire school community. In this way, the SchoolAIR framework may have a relevant role in increasing the sustainability and the resilience of school environments by enhancing ventilation procedures adapted to deal with high outdoor pollution levels.

The SchoolAIR framework can be adapted to integrate the monitoring of other types of pollutants in the case of schools located near specific sources, such as refineries, airports or other types of infrastructure that emit some type of specific pollutant that could compromise the health of the occupants of the school building (e.g., benzene, volatile organic compounds, formaldehyde or nitrogen dioxide).

The next step of this process is to extend the application of the SchoolAIR framework to other schools. As future research, the SchoolAIR framework can be used to analyse how different classroom conditions and different ventilation procedures influence the recovery of IAQ to adequate levels. In this scope, the analysis of a larger data-set, including outdoor/indoor conditions for different environments, building structures and ventilation practices, has relevant scientific value and could be used to develop a decision-support system for ventilation practices that assure adequate IAQ conditions in school environments.

Future research could also focus on how to use both field data and simulation data to develop predictive AI models: short-term forecast of outdoor air quality conditions would trigger ventilation management protective measures, while long-term forecast would be useful to manage future building infrastructure and guide reference occupancy levels.

## Figures and Tables

**Figure 1 sensors-24-00148-f001:**
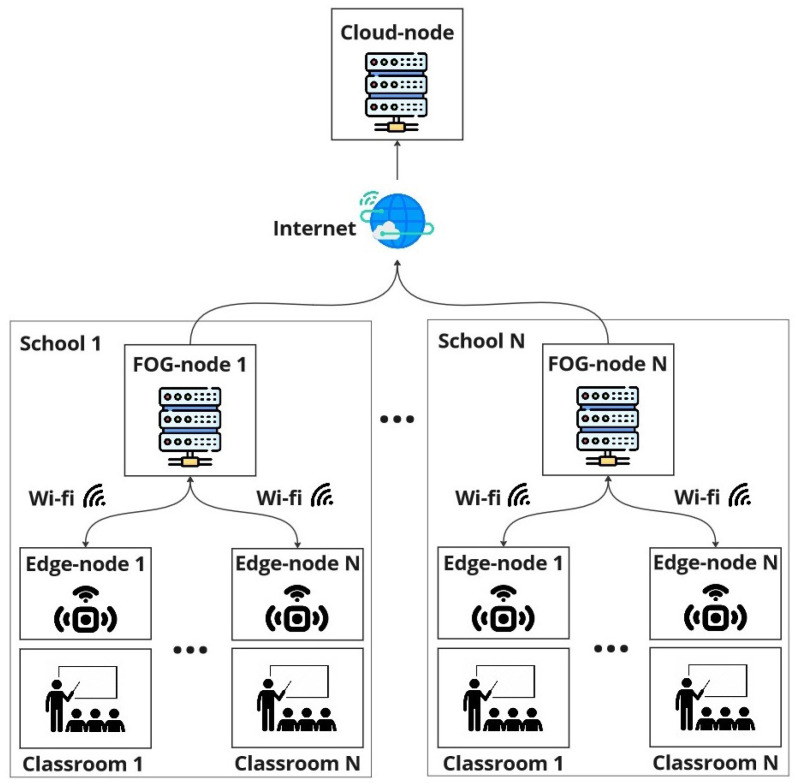
SchoolAIR system’s architecture.

**Figure 2 sensors-24-00148-f002:**
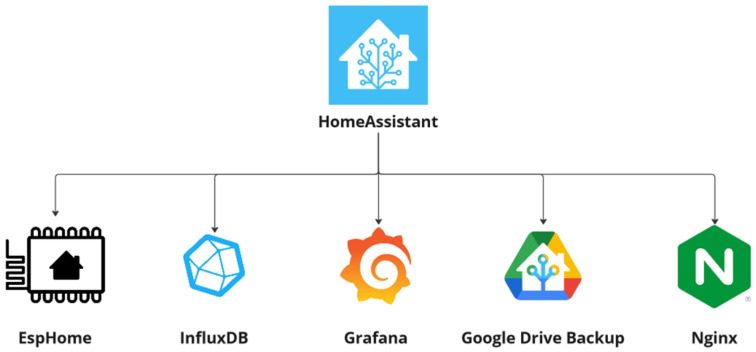
HomeAssistant Integrations used in the SchoolAIR framework.

**Figure 3 sensors-24-00148-f003:**
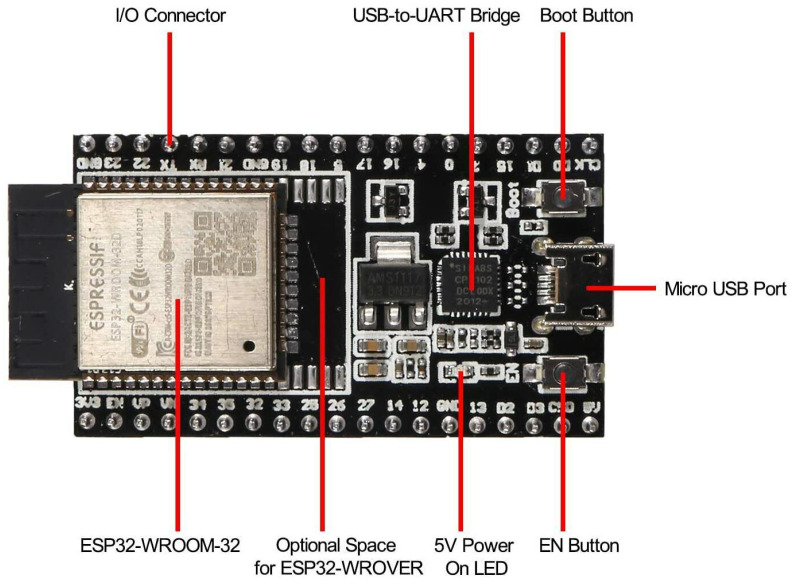
ESP32 board used in Edge-Nodes.

**Figure 4 sensors-24-00148-f004:**
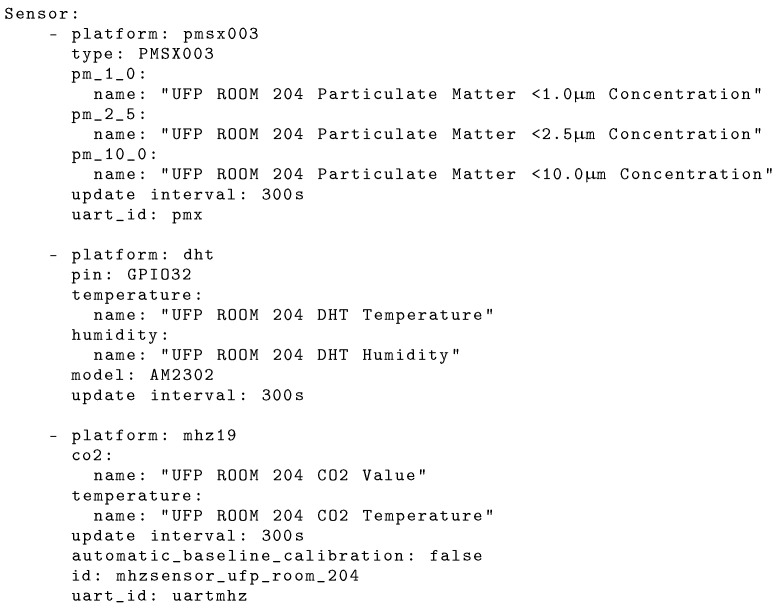
EspHome configuration for an Edge-Node/Mote.

**Figure 5 sensors-24-00148-f005:**
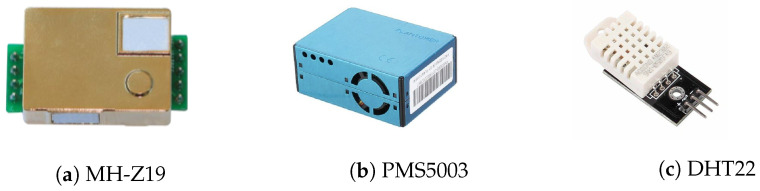
Sensors assembled in the the Edge-Nodes.

**Figure 6 sensors-24-00148-f006:**
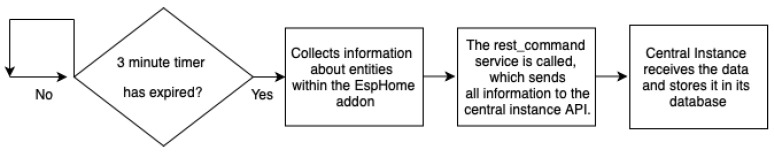
FOG-node data upload cycle.

**Figure 7 sensors-24-00148-f007:**
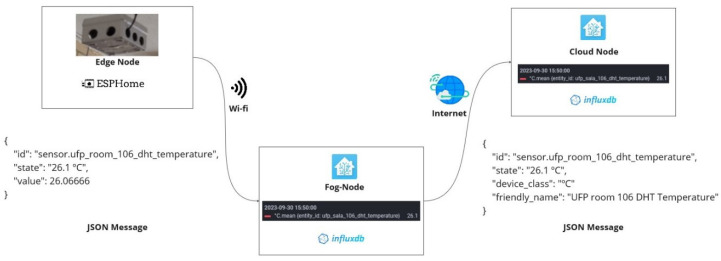
Data flow between Edge, Fog and Cloud Nodes.

**Figure 8 sensors-24-00148-f008:**
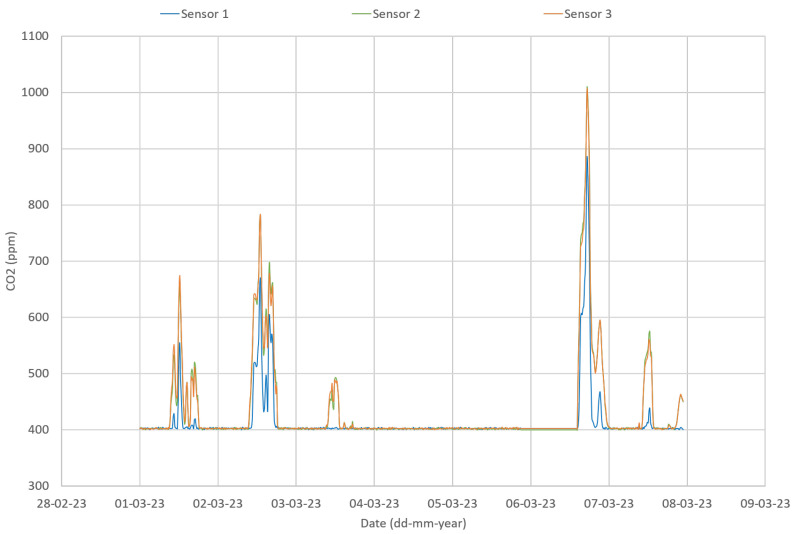
CO_2_ data during sensors inter-comparison at University Fernando Pessoa.

**Figure 9 sensors-24-00148-f009:**
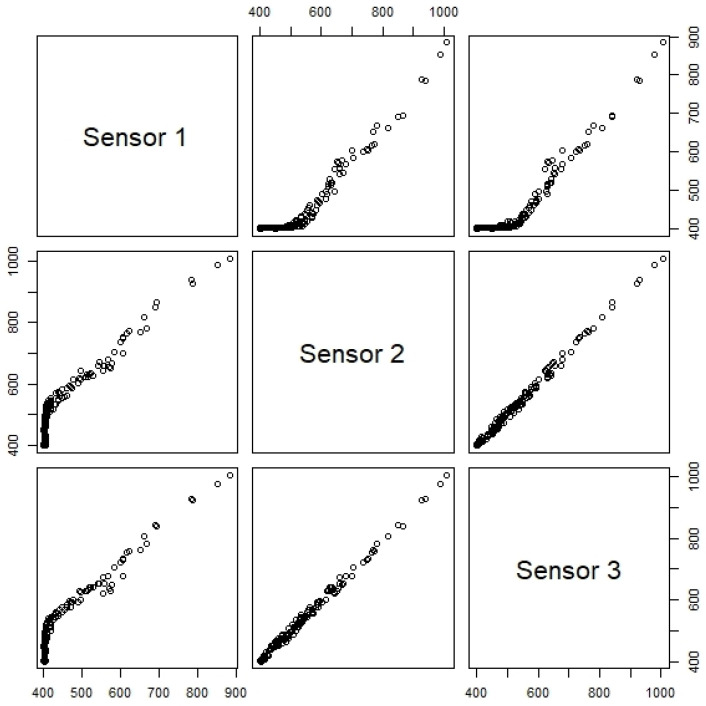
Sensors correlation plots at University Fernando Pessoa for CO_2_ (concentration in ppm).

**Figure 10 sensors-24-00148-f010:**
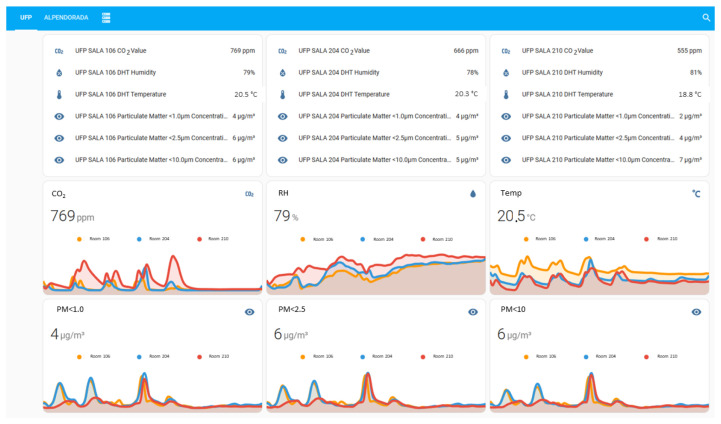
Dashboard with sensors’ monitoring data at University Fernando Pessoa (example of a weekly cycle—week of 10 to 16 April 2023).

**Figure 11 sensors-24-00148-f011:**
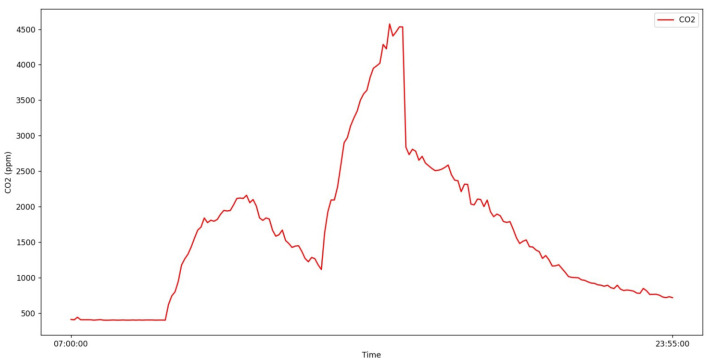
CO_2_ concentration (ppm) in classroom 106 at University Fernando Pessoa—12 April 2023.

**Figure 12 sensors-24-00148-f012:**
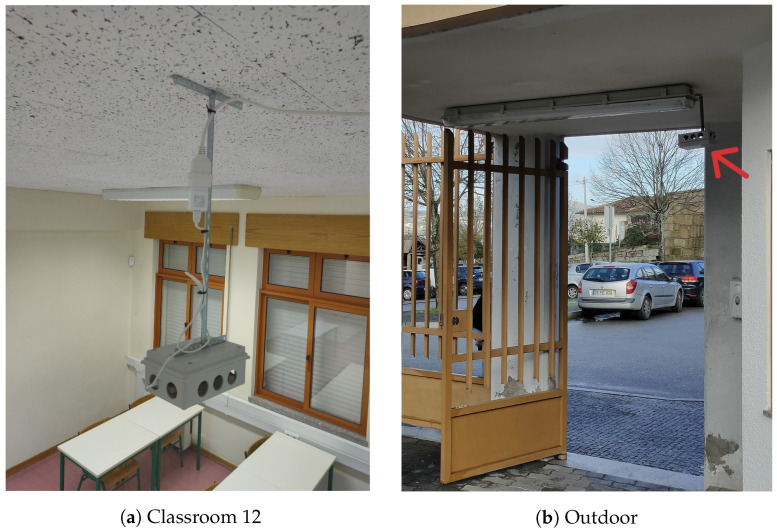
Sensing Edge-Nodes at Alpendorada high school.

**Figure 13 sensors-24-00148-f013:**
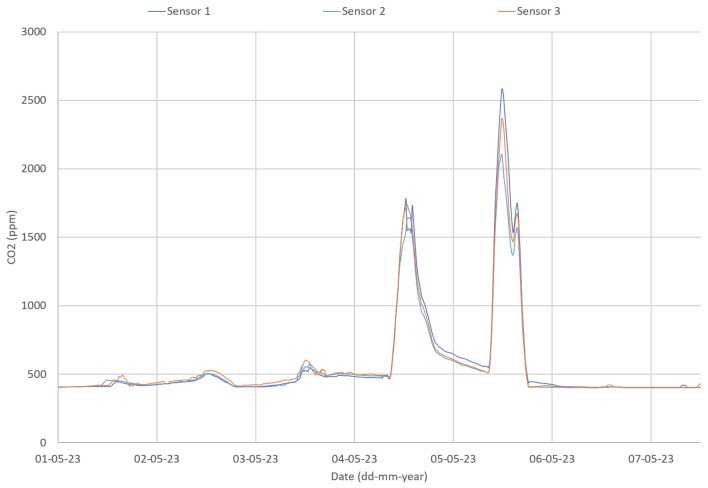
CO_2_ data during sensors inter-comparison at Alpendorada High School.

**Figure 14 sensors-24-00148-f014:**
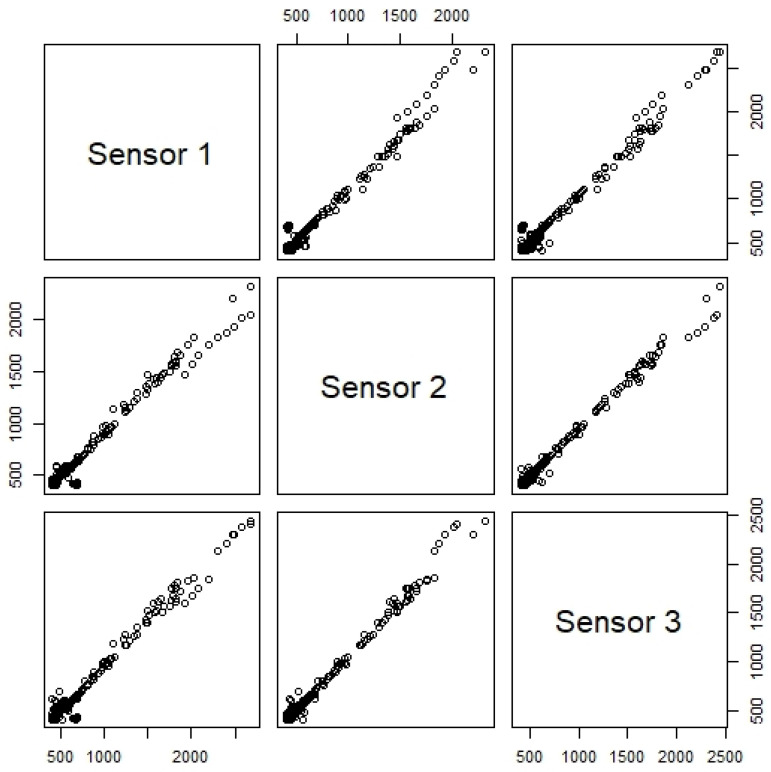
Sensors correlation plots at Alpendorada High School for CO_2_ (concentration in ppm).

**Figure 15 sensors-24-00148-f015:**
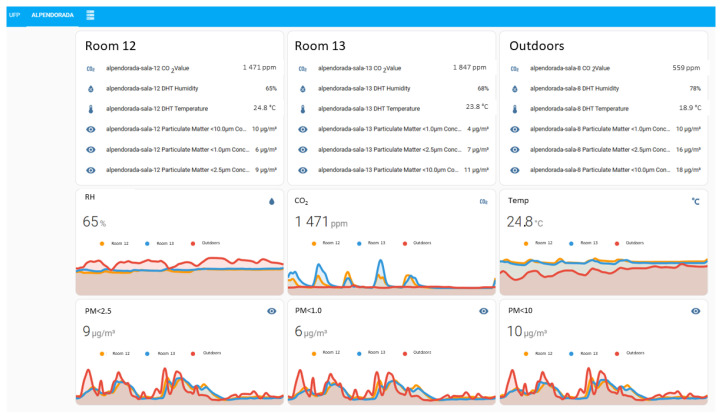
Dashboard with sensors’ monitoring data at Alpendorada high school—week of 5 to 11 November 2023.

**Table 1 sensors-24-00148-t001:** Sensor parameters.

Sensor Name	Size (mm) (L × W × H)	Measurement Range	Resolution and Accuracy	Response Time
MHZ-19	33 × 20 × 9	400 to 5000 ppm	±50 ppm + 3% reading value	T90 < 120 s
PMS5003	50 × 38 × 21	0 to 500 (µg·m^−3^)	±10%	<10 s
DHT22	15.3 × 7.8 × 25.3	−40 to 80 °C 0–100% HR	±0.5 °C ±2% HR	2 s

## Data Availability

The data-sets generated and analysed during the current study are available from the corresponding author on reasonable request.

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
