# Peer review of "SchoolAIR: A Citizen Science IoT Framework Using Low-Cost Sensing for Indoor Air Quality Management"

_sensors, 2023, doi:10.3390/s24010148_

Round 1

Reviewer 1 Report

Comments and Suggestions for Authors

1. Line 8: How does your study improve the technical skills of high-school students? Did you do some studies in this direction? If installing a device means improvement of technical skills then you need to detail more in how exactly the device is mounted or configured by an average person. Not enough information to claim this.

2. Line 63: add the abbreviation for SDG (maybe at line 52)

3. I do not see the relevance of Table 1  and Table 2. Details can be included around Figure 1. Figure 1 already offers a clear understanding on what these tables present. Entire 2.1 subchapter can be omitted.

4. Lines 148-150: These line offers information about fault tolerance and security of the framework. Although there are some references to such approaches is not very clear if the framework really accomplishes this? There are slight references to encryption and redundancy below in the paper. This part should clearly state: which encryption is used, what happens if someone intercepted the data, what happens if some sensors start reporting bad data, how is this mitigated?

5. Line 177 which integrations. Some references are placed below in the paper but you can also state it here as an enumeration. 

6. Lini 212: Assumption that the sensors offer real data with no calibration. You stated that the sensors "make it possible to capture precise" information. It is well known that low-cost sensors have bad sensitivity and do not offer precise information without calibration procedures. Please rephrase.

7. line 217-220 The tables present the same information; they can all be merge in a single table.

8.  Line 243 - PMS is not able to measure pollen. Can you include a reference to this? Pollen usually is larger than 10um.

9. Lines 250-254 - Please add reference to sustain these phrases.

10. Line 324: The procedure may lead to determine faulty sensors but it is not clear that sensor 1 offers bad readings or just have a smaller sensitivity. Figure 7 shows visible correlation between sensors, even sensor 1 follows the CO2 concentrations but with lower sensitivity. It is not always possible to use this testing on a real world scenario. What happens if 10 sensors have different sensitivities (which is most likely to have), will you disregard all of them? Is it not better to use a simple calibration technique regarding CO2? 

11. Figure 9 - add a scale on the X-axis. The period is not clear, although it is stated in the text there is a week, but for example if we look ar RH alone it is not clear in which day the measurements are done.

12. Line 379 - What is the relevance to measure outdoor concentrations for this study? It is clear that outdoor AQI influences the IAQI, but is not very clear in this study how it does so. Please state clearly the importance of outdoor monitoring.

13. Figure 10 does not present the position of the sensor node outdoors. Only a description in the text is used. It would be clearer to see the entire setup.

14. Figure 11 shows a week of measurements but only in one day the CO2 is higher. This is in contradiction with the next week data  presented in Figure 13 where there are clear variations. Was the initial test done in another room? Was it placed in some kind of vacation period? 

15. Overall there were mentions of a low-cost system developed, yet there is no price range of developing, install, and maintain the system. Please state it clearly if this is cheaper how so? Are there any correlation of other low cost sensors instalments? 

16.  The system presents correlations between other sensor nodes. How do the measured values correlates with a reference device? Is the system monitoring real values or just some thresholds? There is no specific information on correlation with reference sensors, or information about the error that the system is prone to. 

17. Line 326: there is no evidence to support this claim. There is no place in the paper that presents the used ventilation techniques or if any particular ventilation can be used with success. If the test used some ventilation techniques that managed to reduce IAQ when a level is exceeded please state it clearly. If no ventilation is used can you conclude that the measured values impacted the students in any way? 

18. Line 432 It is not clear how the outdoor pollution influenced IAQ. In Figure 13 PM measurements fluctuates drastically outdoors but the PM indoors does not necessary follow the trend. Maybe because there is no reference on the ventilation situation, the windows may be closed thus the outside does not influence the inside air. RH, T and CO2 measured outside seem to be stable, no clear indication of how they affect the indoor in the presented test.

19. Lastly what does IAQ really mean in this paper? There are different indexes to represent this but no one is clearly used. If CO2 exceeds some levels does not particularly mean the IAQ is bad. It is not clear in the executed tests if any of the measured parameters really exceeded some certified levels. 

Author Response

Dear reviewer,

The revised version of our manuscript entitled "SchoolAIR: A Citizen Science IoT Framework using Low-Cost Sensing for Indoor Air Quality Management" (manuscript ID: sensors-2766237) has been uploaded. 
We would like to thank you for taking the time to review our work. Your comments were a valuable contribution to its improvement. We agree with most of your comments and have made changes accordingly. The detailed, point-by-point responses to the comments are attached to this letter (Round1_Reviewer_1.pdf).
We believe that all comments have been adequately answered and addressed in this revised version of the manuscript.

Yours sincerely

Nelson Barros

Reviewer 2 Report

Comments and Suggestions for Authors

I would first congratulate the author for their initiative and their work. The paper is very interesting and describe a nice sensing device that uses numeric tools that are commonly used when working on open source. However, I’m concerned about the choice of the CO2 sensor when looking at its measurement range. Taking into consideration the minimum and the accuracy the device seems to not being able to give data below 350ppm. My doubts are based on whether very low concentrations are really taken into account, due to lack of calibration or drift over time. These doubts are particularly supported by figure 7 and 8 where typical false low value are visible at low concentration. However, if the devices are used in a qualitative way (looking at trend for examples) and not a quantitative way (looking at a precise value, data could be filtered.

Comments on the Quality of English Language

No particular comment, only some common error that could be corrected with a careful reading (exple in table 2: "It must assembled from Low-Cost components", do yo mean "It must be assembled from Low-Cost components"; or the definition of MQTT for non specialist readers).

Author Response

Dear reviewer,

The revised version of our manuscript entitled "SchoolAIR: A Citizen Science IoT Framework using Low-Cost Sensing for Indoor Air Quality Management" (manuscript ID: sensors-2766237) has been uploaded. 
We would like to thank you for taking the time to review our work. Your comments were a valuable contribution to its improvement. We agree with most of your comments and have made changes accordingly. The detailed, point-by-point responses to the comments are attached to this letter (Round1_Reviewer_2.pdf).
We believe that all comments have been adequately answered and addressed in this revised version of the manuscript.

Yours sincerely

Nelson Barros

Reviewer 3 Report

Comments and Suggestions for Authors

An multidisciplinary research team has developed a framework - SchoolAIR, based on a low-cost and scalable IoT system architecture to support the improvement of indoor air quality (IAQ) in schools and develop adequate ventilation management. It is shown that students’ technical skills are enhanced since the SchoolAIR framework is based on Do-It-Yourself sensors that continuously monitor air temperature, relative humidity, concentrations of carbon dioxide and particulate matter in school environments. The sensors were calibrated at the University Fernando Pessoa and inter-compared at Alpendorada high school. Detected high values of CO2 indicate the need to introduce fresh air into the classroom through ventilation. Further, the measurement and data analyses results as e.g. that outdoor air is a relevant source of particulate matter demonstrate the importance of real-time monitoring of IAQ and outdoor air pollution levels to support decision-making in ventilation management and assure adequate IAQ.

General comments

It is concluded that the proposed approach encourages the transfer of scientific knowledge from universities to society in a dynamic and active process of social responsibility based on a citizen science approach, promoting scientific literacy of the younger generation and enhancing healthier, resilient and sustainable indoor environments. This is mainly due to the common practice of air exchange in class rooms by opening of windows and doors so that outdoor air pollutants can be transported indoor. Some outlook is given as e. g. that the SchoolAIR framework can be adapted to integrate the monitoring of other types of pollutants in the case of schools located near specific sources.

The paper addresses relevant scientific questions. The paper presents some novel concepts, ideas and tools.

The scientific methods and assumptions are valid and clearly outlined so that substantial conclusions can be reached.

The description of experiments and calculations are sufficiently complete and precise to allow their reproduction by fellow scientists.

The quality of the figures is fine.

The related work is not well cited.

Title and abstract reflect the whole content of the paper. The abstract should include more results and less introduction.

The overall presentation is well structured and clear. The language is fine in detail.

The mathematical formulae, symbols, abbreviations, and units are correctly defined.

Specific Comments

In Fig. 8 the units at the axis are missing.

Technical corrections

The internet addresses of the references are not complete.

Author Response

Dear reviewer,

The revised version of our manuscript entitled "SchoolAIR: A Citizen Science IoT Framework using Low-Cost Sensing for Indoor Air Quality Management" (manuscript ID: sensors-2766237) has been uploaded. 
We would like to thank you for taking the time to review our work. Your comments were a valuable contribution to its improvement. We agree with most of your comments and have made changes accordingly. The detailed, point-by-point responses to the comments are attached to this letter (Round1_Reviewer_3.pdf).
We believe that all comments have been adequately answered and addressed in this revised version of the manuscript.

Yours sincerely

Nelson Barros

Round 2

Reviewer 1 Report

Comments and Suggestions for Authors

Thank you for the clarifications.

I do believe you addressed them as expected.

The work present a new approach, even if in my opinion the quality of the measurements is still not there. Sensor calibration needs to be done at some degree.  The project can increase awareness to AQ and it would be great to see more instalments of your proposed station.

Author Response

Dear Reviewer, 

Once again, thank you very much for taking the time to review our work. Your comments were a very valuable contribution to improving our manuscript.

One last comment on the issue of sensor calibration. In fact, the periodic calibration is an improvement opportunity that will be considered in the SchoolAIR framework.

Thank you!

Best regards,

Nelson Barros